# The Expression and Polymorphism of Entry Machinery for COVID-19 in Human: Juxtaposing Population Groups, Gender, and Different Tissues

**DOI:** 10.3390/ijerph17103433

**Published:** 2020-05-14

**Authors:** Behrooz Darbani

**Affiliations:** The Novo Nordisk Foundation Center for Biosustainability, Technical University of Denmark, 2800 Kongens Lyngby, Denmark; behdas@biosustain.dtu.dk or behroozdarbani@gmail.com; Tel.: +45-53578055

**Keywords:** COVID-19, gender, host entry machinery, polymorphism

## Abstract

(1) *Background*: Combating viral disease outbreaks has doubtlessly been one of the major public health challenges for the 21st century. (2) *Methods*: The host entry machinery required for COVID-19 (SARS-CoV-2) infection was examined for the gene expression profiles and polymorphism. (3) *Results*: Lung, kidney, small intestine, and salivary glands were among the tissues which expressed the entry machinery coding genes *Ace*2, *Tmprss*2, *Cts*B, and *Cts*L. The genes had no significant expression changes between males and females. The four human population groups of Europeans, Africans, Asians, and Americans had specific and also a common pool of rare variants for the X-linked locus of ACE2 receptor. Several specific and common ACE2 variants including S19P, I21T/V, E23K, A25T, K26R, T27A, E35D/K, E37K, Y50F, N51D/S, M62V, N64K, K68E, F72V, E75G, M82I, T92I, Q102P, G220S, H239Q, G326E, E329G, G352V, D355N, H378R, Q388L, P389H, E467K, H505R, R514G/*, and Y515C were of the utmost importance to the viral entry and infection. The variants of S19P, I21T, K26R, T27A, E37K, N51D, N64K, K68E, F72V, M82I, G326E, H378R, Q388L, and P389H also had significant differences in frequencies among the population groups. Most interestingly, the analyses revealed that more than half of the variants can exist in males, i.e., as hemizygous. (4) *Conclusions*: The rare variants of human ACE2 seem to be one of the determinant factors associated with fitness in the battle against SARS viruses. The hemizygous viral-entry booster variants of ACE2 describe the higher SARS-CoV-2 mortality rate in males. This is also supported by the lack of gender bias for the gene expression profiles of entry machinery. A personalized medicine strategy is conceived for isolating high-risk individuals in epidemic circumstances.

## 1. Introduction

The early 21st century has been a momentous era in contagious disease history. During the last twenty years, public health has been repeatedly threatened by the pathogenic respiratory beta-coronaviruses (CoV) including the Severe Acute Respiratory Syndrome CoV (SARS-CoV) [1], Middle East Respiratory Syndrome CoV (MERS-CoV) [2], and very recently by the pandemic burst of SARS-CoV-2 [3]. These are enveloped viruses carrying capped and polyadenylated positive-strand RNA genomes with ∼ 30,000 nucleotides [4]. They have been considered as extraordinary threats to public health by virtue of the progressive respiratory failure and human mortality rates of 4%−36% [5,6,7]. The current SARS-CoV-2 pandemic [3] has resulted in a global crisis after a three-month period of its outset. The mortality rate has also reached to 6.8% measured among 2,397,216 confirmed cases as of April 21st, 2020 [8]. Therefore, the respiratory syndrome by coronaviruses calls for sustainable preventive approaches and effective treatment protocols to combat current and also future outbreaks.

The variations of susceptibility to SARS-CoV among different species have been found associated with the inter-species genetic polymorphism of the host machinery for viral entry [9,10,11]. Also, the polymorphism of entry machinery has linkages with cardiovascular disease, hypertension, and diabetes [12,13,14], which, as comorbidities, significantly increase the severity of SARS-CoV-2 infection [15]. This calls for genetic analysis of the entry machinery in humans to investigate its possible linkage with the individual differences in susceptibility to SARS-CoV-2. This will advance our understanding of mechanisms the virus requires for entry and within-host spread. It will also facilitate the identification of susceptible individuals or groups and the development of treatment strategies.

## 2. Materials and Methods

Gene expression data was extracted from the Genotype-Tissue Expression (GTEx) project [16] publicly available at https://www.gtexportal.org/home/. Tissue-specific co-expression analysis was performed on TPM measurements by using the BicMix biclustering. The human SNP data was extracted from the GenBank, the database for Single Nucleotide Polymorphisms (dbSNP) including the 1000 genomes project data, the exome aggregation consortium data, and the genome aggregation data. For every variant, data was obtained from the experiments and pooled. Data, i.e., number of individuals carrying the reference or variant alleles, was processed using chi-square statistics. The abundance of the rare variant (sum of individuals with rare variant) and the corresponding reference allele (sum of individuals with reference allele) in every comparison were used to calculate the expected values for the chi-square test.

## 3. Results and Discussion

The genetic diversity and expression of the host factors required for SARS-CoV/CoV-2 entry were explored among the human population groups and tissues, respectively. The angiotensin-converting enzyme 2 (ACE2) [17,18], the endosomal cysteine protease cathepsin B (CTSB) and L (CTSL) [18,19], and the transmembrane serine protease 2 (TMPRSS2) [18,20,21] are exploited by SARS-CoV/CoV-2 as the cellular-entry machinery. The Genotype-Tissue Expression (GTEx) data [16] was used to inspect the tissue expression profiles. The *Ace*2, *Tmprss*2, *Cts*B, and *Cts*L genes were found expressed in different tissues (Figure 1a). The minor salivary glands, lung, small intestine, liver, kidney, and heart were among the tissues in which the genes were expressed (Figure 1a). *Ace*2 and *Tmprss*2 showed no considerable expression in the brain and spleen as *Tmprss*2 in the heart (Figure 1a). Among the three cysteine and serine protease coding genes, *Tmprss*2 had the highest correlated expression with *Ace*2 (Figure 1b). Furthermore, lung, kidney, small intestine, and salivary glands were co-clustered together within a group of tissues with the highest expression levels by considering all of the four genes (Figure 1b). The additive effects of the co-expressed cysteine and serine proteases on the viral entry suggest targeting of the TMPRSS2, CTSB, and CTSL proteases simultaneously, otherwise the ACE2 receptor for a robust treatment.

By inspecting the human genetic variants pool available at https://www.ncbi.nlm.nih.gov/snp/, ∼ 100 to 400 missense SNPs were extracted after filtering for the non-coding regions of *Ace*2, *Tmprss*2, *Cts*B, and *Cts*L genes (Table 1). In contrast to the two alleles of *Tmprss*2, *Cts*B, and *Cts*L present in both the males’ and females’ genomes, there is one allele for the X-linked *Ace*2 locus in males’ genomes (Table 1). As the host receptor, ACE2 interacts with the receptor binding motif (RBM) of the viral S1 protein which is essential for SARS-CoV/CoV-2 infection [17,18,22]. Further analyses were executed on the *Ace*2 variants to compare the human population groups of Europeans, Africans, Asians, and Americans. The *Ace*2 locus had 265 missense SNPs, including inframe insertions and deletions, which were distributed in the exons I to XIII, XVII, and XVIII (Table 1, Figure 2a). Of these, 194 SNPs were found with allelic frequencies by considering the 1000 genomes project data, the exome aggregation consortium data, and the genome aggregation data (Figure 2a).

The global prevalence for rare variants of *Ace*2 ranged from 0.00001 to 0.016 and had an average and median of 2.14 × 10^−04^ and 3.59 × 10^−05^, respectively. Several SNPs had significant frequency-changes, at least in one of the population comparisons (Figure 2a). Statistically significant population-specific SNPs were also detected, which was more pronounced for Africans (Table 2). Interestingly, the amino acid substitutions in several variants can potentially influence the interaction between the ACE2 and the viral S1 protein, and thereby the viral infectivity. Different amino acid residues distributed all over the ACE2 receptor, have been found very influential either by facilitating or by hindering the viral infectivity [22,23,24]. Using this knowledge, 13 variants (rs73635825 [S19P], rs778030746 [I21V], rs1244687367 [I21T], rs756231991 [E23K], rs1434130600 [A25T], rs4646116 [K26R], rs781255386 [T27A], rs778500138 [E35D], rs1199100713 [N64K], rs867318181 [E75G], rs763395248 [T92I], rs1395878099 [Q102P], and rs142984500 [H378R]) were found as the interaction-booster between ACE2 and S1. The amino acid exchanges at the positions of 25, 35, and 75 had no reported frequency. The abundance of S19P, I21T, K26R, T27A, N64K, and H378R variants significantly differed among the population groups (Figure 2a). Interestingly, H378R and S19P were Europeans (Figure 2a, Table 2, *p* < 0.0449, frequency: 0.00014) and Africans (Figure 2a, Table 2, *p* < 0.0000, frequency: 0.0033) specific variants, respectively. Furthermore, another group of 18 SNPs including rs1348114695 [E35K], rs146676783 [E37K], rs1192192618 [Y50F], rs760159085 [N51D], rs1569243690 [N51S], rs1325542104 [M62V], rs755691167 [K68E], rs1256007252 [F72V], rs766996587 [M82I], rs759579097 [G326E], rs143936283 [E329G], rs370610075 [G352V], rs961360700 [D355N], rs751572714 [Q388L], rs762890235 [P389H], rs1016409802 [H505R], rs1352194082 [R514G/*], and rs1263424292 [Y515C] were found as interaction-inhibitor variants. The amino acid substitutions at the positions 505, 514, and 515 had no reported frequency. The variants E37K, N51D, K68E, F72V, M82I, G326E, Q388L, and P389H had significant changes in frequency among the population groups (Figure 2a). The Q388L and M82I were also found as Americans (Figure 2a, Table 2, *p <* 0.0345, frequency: 0.00016) and Africans (Figure 2a, Table 2, *p* < 0.0066, frequency: 0.00021) specific variants, respectively. The impact on the viral infectivity was not clear for the Asians specific variants of rs751603885 [R697G], rs763593286 [A532T], rs745514718 [S257N], and rs200180615 [E668K].

As an X-linked phenotype, the efficacy of interaction-booster and interaction-inhibitor rare variants of ACE2 can be stronger in males than females. This is due to the hemizygosity in males. To examine whether the rare variants of the X-linked *Ace*2 locus are lethal and have been under strong negative selection or not, the incidence of variants in males, i.e., the hemizygous state, were analyzed using the genome aggregation data (https://gnomad.broadinstitute.org/). Here, the majority (98.3%) of missense variants were as heterozygous in females. In addition, 51% of the variants were also present in males indicating non-lethality in the hemizygous state. Finally, 7 out of 10 interaction-booster variants (S19P, I21V, I21T, K26R, N64K, T92I, and H378R) and 9 out of 14 interaction-inhibitor variants (E35K, E37K, N51D, M62V, K68E, F72V, E329G, Q388L, and P389H) were also found in males. Therefore, more than half of the variants have the chance to influence the interaction between the human ACE2 and the viral S1 protein in the hemizygous state, i.e., in the absence of the reference allele, in males. Accordingly, gender bias has been observed towards a higher mortality rate in males accounting for 60%−70% of death despite the similar SARS-CoV-2 infection rates between the genders [5,25].

Weak ACE2-S1 interactions lead to lower levels of SARS-CoV susceptibilities in rats, and also to some extent, in pig and mouse [9,10]. Three additional SNPs including rs774621083, rs1448326240, and rs1270795706, all perceived as interaction-inhibitor variants, were further selected by juxtaposing the human, rat, mouse, and pig ACE2s (Figure 2b). The variant rs774621083 had a polar serine residue at position 220 instead of the non-polar glycine (Figure 2b). The pig, mouse, and rat ACE2s have the polar amino acid asparagine at this position (Figure 2b). This variant had a global abundance of 2.9 × 10^−5^ without significant difference among the population groups (Figure 2a). As seen in rat, rs1270795706 had the positively charged residue of lysine at the position 467 and the Europeans specific variant rs1448326240 had the polar residue of glutamine at the position 239 in replacement of the negatively charged glutamic acid and the positively charged histidine, respectively (Figure 2a, b).

## 4. Conclusions

Taken together, the human ACE2 has a rich pool of rare variants, which can explain the individual competence in the battle against the SARS viruses. Most interestingly, there are statistically significant variations in the frequencies of the rare variants among the human population groups. These alleles, as 34 introduced in this study, can potentially be decisive in SARS-CoV/CoV-2 recognition and infection. As an X chromosome linked phenotype, more than half of the ACE2 rare variants were also found in males. The impact of the ACE2 rare variants can be more substantial in males than females. Accordingly, the interaction-booster variants of ACE2 can explain the gender bias towards a higher mortality rate caused by SARS-CoV-2 in males; this is in line with the lack of gender bias for the expression of *Ace*2, *Tmprss*2, *Cts*B, and *Cts*L in asexual tissues. For any population group of interest, therefore, it is worth to investigate the enrichment of the rare variants among the SARS-CoV-2 infected cases with severe symptoms. The results have the potential to advance the personalized medicine strategies, e.g., by screening for the high-risk individuals that need isolation against the viral disease outbreaks.

## Figures and Tables

**Figure 1 ijerph-17-03433-f001:**
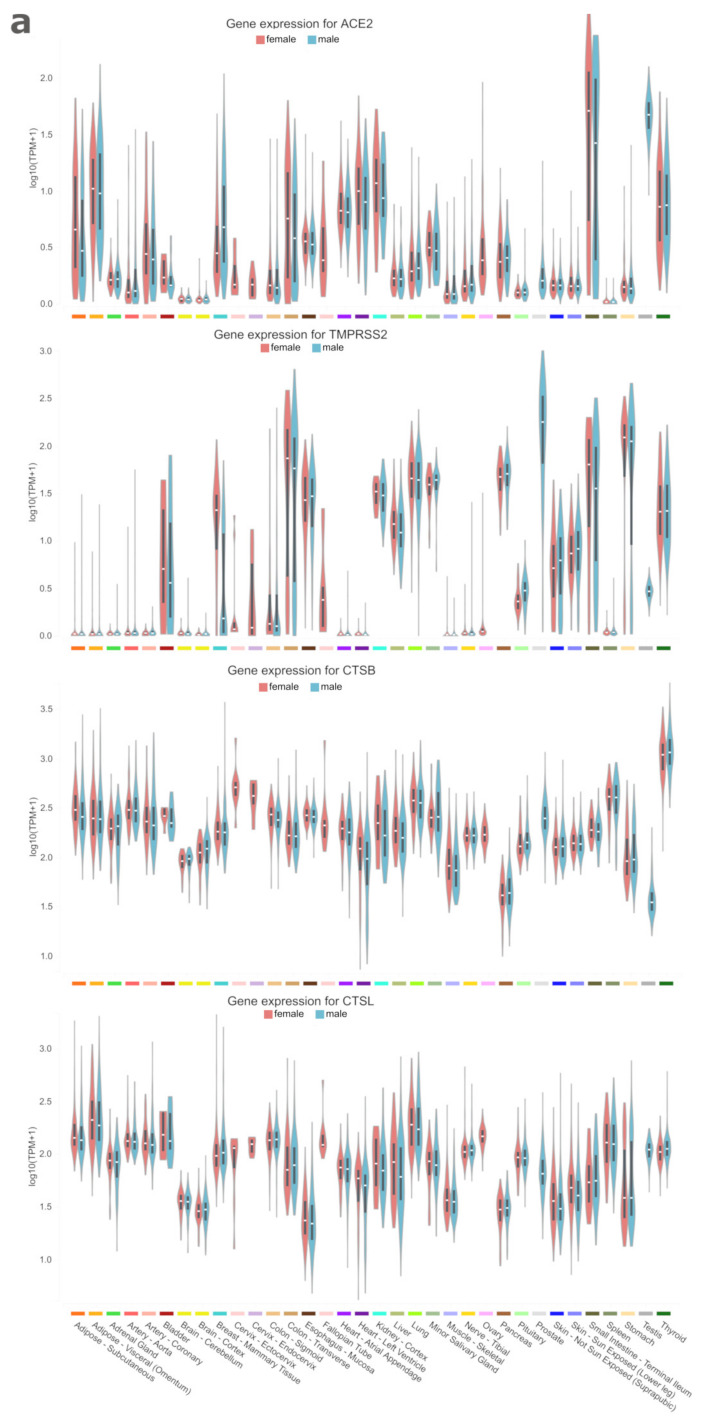
Gene expression profiles for *Ace*2, *Tmprss*2, *Cts*B, and *Cts*L. (**a)**, Expression profiles across different tissues. Boxplots are shown as median and 25^th^−75^th^ percentiles. (**b**), Tissue-specific co-expression analysis on TPM measurements by using the BicMix biclustering. (**a**,**b**), Data extracted from the Genotype-Tissue Expression (GTEx) project [16]. TPM: transcripts per million.

**Figure 2 ijerph-17-03433-f002:**
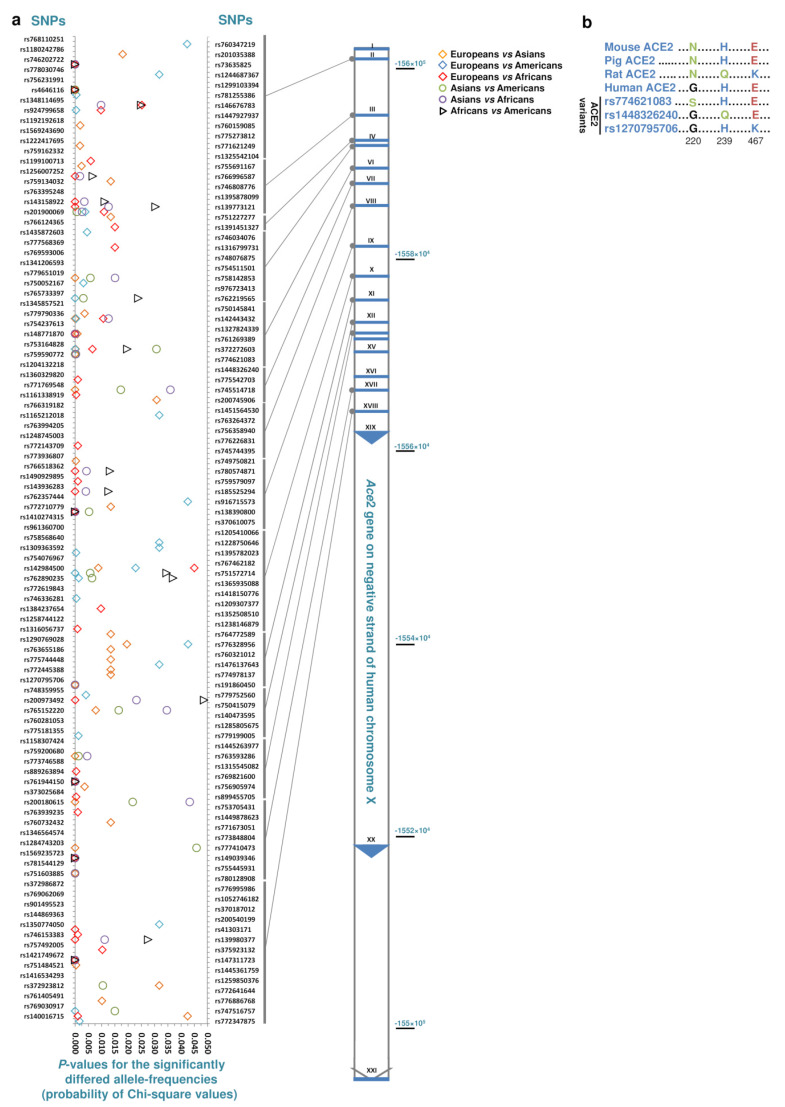
The rare missense variants of ACE2. (**a**), SNPs are demonstrated with their respective chromosomal positions. Data, i.e., number of individuals carrying the reference or rare alleles, was processed using the chi-square statistics. The probability for chi-square values are only shown for the significant events. Data extracted from the GenBank, the database for Single Nucleotide Polymorphisms (dbSNP) including the 1000 genomes project data, the exome aggregation consortium data, and the genome aggregation data. I-XXI: Exons for *Ace*2. (**b**), Additional missense mutations in the human ACE2 assumed to interfere with the viral S1 protein interaction.

**Table 1 ijerph-17-03433-t001:** Chromosomal location and Single Nucleotide Polymorphism (SNP) characteristics of the human genes required for SARS-Cov/CoV-2 entry.

Genes	*Ace*2	*Tmprss*2	*Cts*B	*Cts*L
Chromosome	X	21	8	9
Intronic-SNPs	15,391	10,293	9470	1101
Synonymous-SNPs	104	173	149	50
Missense SNPs, inframe insertion/deletion SNPs	265	394	311	102

Data extracted from the GenBank, database for Single Nucleotide Polymorphisms (dbSNP) at https://www.ncbi.nlm.nih.gov/snp/.

**Table 2 ijerph-17-03433-t002:** Number of SNPs for the X-linked *Ace*2 locus with significant variations in abundance.

Comparisons	Total	Specific for the Common Population of the Comparisons		
		Specific SNPs in Either of Comparisons	Specific SNPs in All of the Comparisons (Frequency)	Population Size (Avg./Median)
Europeans vs. Asians, Americans or Africans	86	12	1 (0.000143)	122,845/147,472
Asians vs. Europeans, Americans or Africans	55	43	4 (0.00013–0.00030)	40,150/45,859
Americans vs. Africans, Asians or Europeans	52	29	2 (0.00016, 0.00018)	29,909/35,302
Africans vs. Americans, Asians or Europeans	47	46	8 (0.00013–0.00336)	20,482/21,424

Data extracted from the GenBank, database for Single Nucleotide Polymorphisms (dbSNP) including the 1000 genomes project data, the exome aggregation consortium data, and the genome aggregation data available at https://www.ncbi.nlm.nih.gov/snp/. Data processed using the chi-square statistics (the frequencies of rare variants and their corresponding reference alleles were used to calculate their expected values in order to execute the test).

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
