# Peer review of "The Expression and Polymorphism of Entry Machinery for COVID-19 in Human: Juxtaposing Population Groups, Gender, and Different Tissues"

_ijerph, 2020, doi:10.3390/ijerph17103433_

Round 1

Reviewer 1 Report

The author has examined diversity of expression and SNPs of the ACE2, TMPRSS2, CTSB and CTSL genes, which regulate SARS-Cov/Cov2 entry, by bioinformatics analysis of the publicly available GTEX and SNP databases of infected populations. The overall objective was to correlate genetic diversity of these genes to the variability of CoVid-19 related morbidity and mortality. Through this metadata analyses, author identified several SNPs with potential impact on virus entry. Although, the conclusion is not very robust in the absence independent supporting data, the result suggest important potential role of the SNPs, especially in ACE2, worthy of future investigation to better understand mechanism of SARS-Cov/Cove2 entry and spread.

However, published reports generally suggested almost equal infection in both male and female Covid-19 patients. Diversity of  X-linked ACE2 gene does not clearly explain the infection similarity in both genders. Effect of gender specific SNP or haplotype is not apparent from the analysis. Thus authors should modify the conclusion and emphasize their findings without associating with much gender bias.

Reviewer 2 Report

This manuscript identifies several ACE2 variants in human population that could help in delivering personalized medicine in COVID-19 patients. Despite the manuscript is well described and the analysis are well conceived, the conclusions reported both in Abstract and in Conclusion sections appears to be quite confusing and not really narrowing to the main findings of the paper. My suggestion is to rephrase these sections accordingly, trying to more specifically summarize and identify key points, and describing how these results could advance current treatments.

Reviewer 3 Report

Dear author,

I have read the manuscript entitled, 'The expression and polymorphism of entry machinery for COVID-19 in human: juxtaposing population groups, gender, and different tissues' with high interest. The concept of the paper is novel, and of high significance. Hence I am recommending to accept this paper after minor revisions.

  1. The figures need to be presented with high clarity and contrast. Figure 1 is very difficult to understand considering the low font size. Please revise it.
  2. Do you assume any significant changes between North American and Latin American population. Have you looked it into?
  3. The introduction is very general and difficult to correlate with the present work. Please revise it for more convenience. 
  4. Overall the manuscript requires minor English and grammatical corrections.
